# Privacy Preserving Face Recognition in Cloud Robotics: A Comparative Study

Chiranjeevi Karri [1], Omar Cheikhrouhou [2,3,*], Ahmed Harbaoui [4], Atef Zaguia [5] and Habib Hamam [6,7]

1  C4-Cloud Computing Competence Centre, University of Beira Interior, 6200-506 Covilhã, Portugal; karri.chiranjeevi@ubi.pt
2  CES Laboratory, National School of Engineers of Sfax, University of Sfax, Sfax 3038, Tunisia
3  Higher Institute of Computer Science of Mahdia, University of Monastir, Monastir 5019, Tunisia
4  Faculty of Computing and Information Technology, King AbdulAziz University, Jeddah 21589, Saudi Arabia; aharbaoui@kau.edu.sa
5  Department of Computer Science, College of Computers and Information Technology, Taif University, P.O. BOX 11099, Taif 21944, Saudi Arabia; zaguia.atef@tu.edu.sa
6  Faculty of Engineering, Université de Moncton, Moncton, NB E1A3E9, Canada; habib.hamam@umoncton.ca
7  Department of Electrical and Electronic Engineering Science, University of Johannesburg, Johannesburg 2006, South Africa
*  Correspondence: omar.cheikhrouhou@isetsf.rnu.tn

**Abstract:** Real-time robotic applications encounter the robot on board resources' limitations. The speed of robot face recognition can be improved by incorporating cloud technology. However, the transmission of data to the cloud servers exposes the data to security and privacy attacks. Therefore, encryption algorithms need to be set up. This paper aims to study the security and performance of potential encryption algorithms and their impact on the deep-learning-based face recognition task's accuracy. To this end, experiments are conducted for robot face recognition through various deep learning algorithms after encrypting the images of the ORL database using cryptography and image-processing based algorithms.

**Keywords:** cloud robotics; image face recognition; deep learning algorithms; security; encryption algorithms





## 1. Introduction

Advancements in the robotics field have led to the emergence of a diversity of robot-based applications and favored the integration of robots in the automation of several applications of our daily life. Multi-robot systems, where several robots collaborate in the achievement of a task [1], are now used in several applications including smart transportation [2], smart healthcare [3], traffic management [4], disaster management [5], and face recognition [6,7]. Although robots' resources have been improving in terms of energy, computation power, and storage, they still cannot satisfy the need of emerging applications [8]. As a solution, researchers focused on solutions that leverage the use of cloud computing [9]. A new paradigm has emerged, namely cloud robotics [8]. Cloud robotics resulted from the integration of advancement in the robotics field with the progress made in the cloud computing field.

Cloud robotics have several advantages compared to traditional robotics systems, including large storage, remote data availability, and more computing power.

The need for computing power is also motivated by the emergence of a new generation of applications using artificial intelligence and learning algorithms to analyze and interpret data. Thanks to these resource powered robotics systems, complex problems that have long been considered very difficult, such as speech and face recognition, can now be executed on robots and have also achieved very promising results. More precisely, it is possible nowadays to design a robot with limited resources and to execute facial recognition tasks

using convolutional neural network (CNN) algorithms by simply connecting to a cloud service [7].

However, this solution faces security problems. Indeed, it is essential to ensure the security of the facial images to be sent through the network. An interesting solution for this problem is the use of a cryptographic system allowing for avoiding network attacks able to recover these data.

In the present work, we focus on the two environments: robot and cloud. Certainly, the confidentiality of robots is essential. This is because private information is shared in public clouds. As a result, there is a clear risk of abuse or at least misuse of private data. The robot contains and uses private information. They should be safeguarded and treated with respect for confidentiality and privacy.

The contribution of this paper is threefold:

- We provide a security analysis of the potential encryption algorithms that can be used to encrypt images stored on the cloud.
- We present a comparative and experimental study of several CNN based secure robotic facial recognition solutions.
- We study the impact of encryption algorithms on the performance of the CNN based robot face recognition models.
  The experimental done in this paper includes several combinations of various encryption algorithms and deep learning algorithms that have been tested and have shown an improvement in recognition speed and accuracy without impacting privacy issues when executed on cloud compared to their execution in robot environment.

The remainder of this paper is structured as follows: Section 2 presents an overview of the main CNN based robot face recognition models. Then, Section 3 highlights the different encryption techniques that can be used for images encryption. Section 4, provides a security analysis of the encryption algorithms studied. The performance of the CNN based robot face recognition and the impact of the encryption algorithms on their performance were presented and discussed in Section 5. Later, The benefit of outsourcing computation to the cloud for face recognition algorithms is shown before concluding the paper.

## 2. CNN Models for Robot Face Recognition

The evaluation of convolution neural network (CNN) tremendously changed the researchers thought process towards the applications of computer vision like object recognition, semantic segmentation, image fusion and so on. It play key role in machine learning or deep learning algorithms. The major difference between these two is in their structure. In machine learning architecture, features are extracted with various CNN layers and classification is done with other classification algorithm whereas in deep-learning both feature extraction and classification are available in same architecture [10]. The artificial neural network (ANN) are feedback networks and CNN are feed-forward networks which are inspired by the process of neurons in human brain. It (ANN's) has mostly one input, output and one hidden layers depends on the problem one can increase the hidden layers. In general, a CNN has a convolution layer, an activation layer, a pooling layer, and a fully connected layer. Convolution layer has a number of filters of different sizes (such as $3 \times 3$, $5 \times 5$, $7 \times 7$) to perform convolution operation on input image aiming to extract the image features. To detect features, these filters are sliding over the image and perform a dot product, and these features are given to an activation layer. In the activation layer, activation function decides the outcome. The main activation functions are: binary step, linear activation, Sigmoid, and Rectified linear unit (ReLu). Especially in our article, we preferred ReLu activation function and its respective neuron outcome becomes one if summed multiplied inputs and weights exceeds certain threshold value or else it becomes zero. In certain region it obeys the linearity rule between input and output of respective neuron. Once features are extracted with various kernels, for dimensionality reduction, outcome of CNN are passed through pooling layer. In present research, there are various ways pool the layer. Average pooling is one simple approach in which average

of feature map is consider, in max pooling, consider maximum among the feature map. Finally, a fully connected layer resulting from the pooled feature map is converted to a single long continuous linear vector as shown in Figure 1. In what follows, we discuss the main CNN models including LeNet, Alexnet, VGG16Net, GoogLeNet, ResNet, DenseNet, MobileFaceNet, EffNet, and ShuffleNet.

Convolutional neural networks (CNN or ConvNet) present a category of deep neural networks, which are most often used in visual image analysis [10]. The model of connectivity between the CNN neurons is inspired from the organization of the animal visual cortex. A CNN is generally composed of a convolution layer, an activation layer, a pooling layer, and a fully connected layer. The convolution layer includes a set of filters of different sizes (e.g., $3 \times 3, 5 \times 5, 7 \times 7$). These filters are applied in a convolution operation on the input image in order to extract the image features. To detect these features, the input image is scanned by these filters and a scalar product is performed. The obtained features presents the input of the activation layer which decides on the outcome.

The main activation functions are: binary step, linear activation, Sigmoid and ReLu. We opted, in this work, for a rectified linear unit transformation function (ReLu). The ReLu transformation function activates a node only if the input is greater than a certain threshold. If the input is less than zero, the output is zero. On the other hand, when the input exceeds a certain threshold, the activation function becomes a linear relationship with the independent variable. Then, the rectified features go through a pooling layer. Pooling is a downsampling operation reducing the dimensionality of the feature map. Pooling can be average pooling, which calculates the average value for each patch on the feature map, or max pooling, which calculates the maximum value for each patch on the feature map. In the last step, a fully connected layer incorporating all features is converted into one single vector, as shown in Figure 1.

Let us now discuss some commonly used CNN models including LeNet, Alexnet, VGG16Net, GoogLeNet, ResNet, DenseNet, MobileFaceNet, EffNet, and   ShuffleNet.

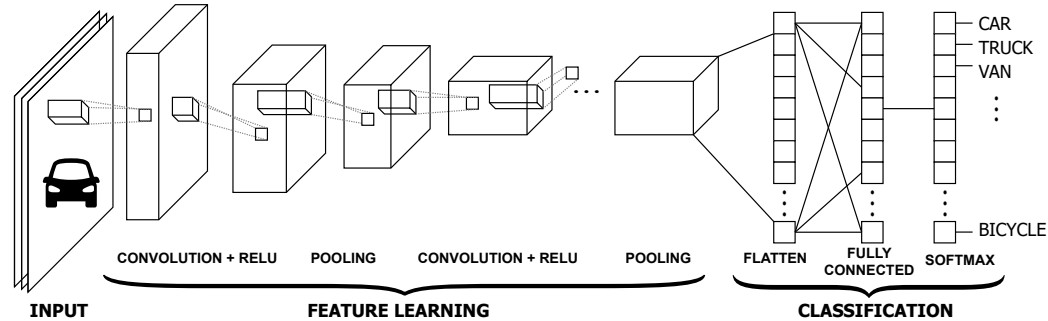

**Figure 1.** CNN architecture.

### 2.1. LeNet

In 1989, the LeNet was introduced by the researcher named Y. LeCun and was used for object recognition especially in images with low resolution. Initially, the input image is pre-processed to size of $3 \times 32 \times 32$ and with six kernels or filter, image features are extracted of size $6 \times 28 \times 28$. These features are passed through average pooling and its outcome is reduced version of original of size $6 \times 14 \times 14$. In next stage of filtering (feature extraction), the same cascaded operations are performed with 16 kernels which leads to size of $6 \times 10 \times 10$. In last stage of feature extraction, two cascaded polling layers applied which leads to size of $16 \times 5 \times 5$ and $120 \times 1 \times 1$ respectively. In classification stage, two fully connected layers (FCN) are used which outcome is of size 4096 [11]. The Figure 2 replicates the hole architecture of LeNet.

### 2.2. AlexNet

AlexNet is a cnn that has had a significant impact on deep learning, particularly in the training and testing process to machine vision. It successfully winning the 2012 ImageNet LSVRC-2012 contest by a significant margin (15.3 percent mistake rates versus 26.2 percent error rates in second position). The network's design was quite similar to that of LeNet, although it was richer, with much more filters per layer and cascaded convolution layers.The AlexNet has eight weighted layers, the first five of which are convolutional and the last three of which are completely linked. The last fully-connected layer's output is sent into a 1000-way softmax, which generates a distribution across the 1000 class labels. The network aims to maximise the multi-variable logistic regression goal, which is really the mean of the log-probability of the right label underneath the forecast distributions throughout all training cases as in Figure 2. Only those kernel mappings in the preceding layer that are on the same GPU are interconnected to the filter or kernels of the 2nd, 4th, and 5th convolutional layers. All kernel/filters mappings in the 2nd layer are connected to the filters of 3rd layer of convolutional. Every neurons throughout the preceding or previous layer are connected to FCN layer of neurons. It supports parallel training on two GPU's because of group convolution compatibility [12]. The architecture of AlexNet is replicated in Figure 2.

### 2.3. Visual Geometry Group (VGG16Net)

This convolution net was invited in 2004 by simonyan and it is available in two forms, one named VGG16 has 16 layers and VGG19 has 19 layers with filter size of $3 \times 3$. It also won the first prize with 93 percent accuracy in training and testing. It takes input image of size (224, 224, 3). The first two layers share the same padding and have 64 channels with 3*3 filter sizes. Following a stride (2, 2) max pool layer, two layers with 256 filter size and filter size convolution layers are added (3, 3). Following that is a stride (2, 2) max pooling layer, which is identical to the preceding layer. There are then two convolution layers with filter sizes of 3 and 3 and a 256 filter. Depends on the requirement of the user one can increase the number of layers for deeper features for better classification Figure 2. It has 138 million hyper-parameters for tuning. It offers parallel processing for reduction in computational time and uses max-pooling which obeys sometime non-linearity [13].

### 2.4. GoogleNet

It is invited and won the prize in ILSVRC in 2014. Its architecture are as similar to other nets In contrast, dropout regularisation is used in the FCN layer, and ReLU activation function is used in all convolution operation. This network, however, is significantly bigger and longer than AlexNet, with 22 overall layers and a far fewer number of hyper-parameters. For computational expanses reduction, in GoogleNet, in-front of $3 \times 3$ and $5 \times 5$ they used $1 \times 1$. This layer is called bottleneck layers as in Figure 3. It optimizes the feature maps based on back propagation algorithm with bearing the increment in computational cost and kernel size. This computational cost is reduced by bottleneck layer which transform the feature maps to matrix of smaller size. This reduction is along the volume direction, so feature map depth becomes less. It has 5 million hyper-parameters and obtained accuracy is near around 93 percent [14].

### 2.5. ResNet

It also own the prize in ILSVRC—2015 and was developed by He et al. In this architecture, performance metric (accuracy) becomes saturated as the features are deeper (number of convolution layers) and after certain extent of deep features its performance is drastically degrading. This architecture offers skip connections for improvement of performance and to get rapid convergence of network. In training phase of the network, these skip connections allows the data to move in another flexible path for better gradient. The ResNet architecture is somewhat complex than VGGNet. It comprises 65 million hyper-parameters and accuracy is near around 96 to 96.4 percent, this value is better than

human accuracy (94.9 percent). A deeper network can be made from a shallow network by copying weights in a shallow network and setting other layers in the deeper network to be identity mapping. This formulation indicates that the deeper model should not produce higher training errors than the shallow counterpart [15]. From Figure 4, ResNet, layers learn a residual mapping with reference to the layer inputs $F(x) := H(x) - x$ rather than directly learning a desired underlying mapping $H(x)$ to ease the training of very deep networks (up to 152 layers). The original mapping is recast into $F(x) + x$ and can be realized by shortcut connections.

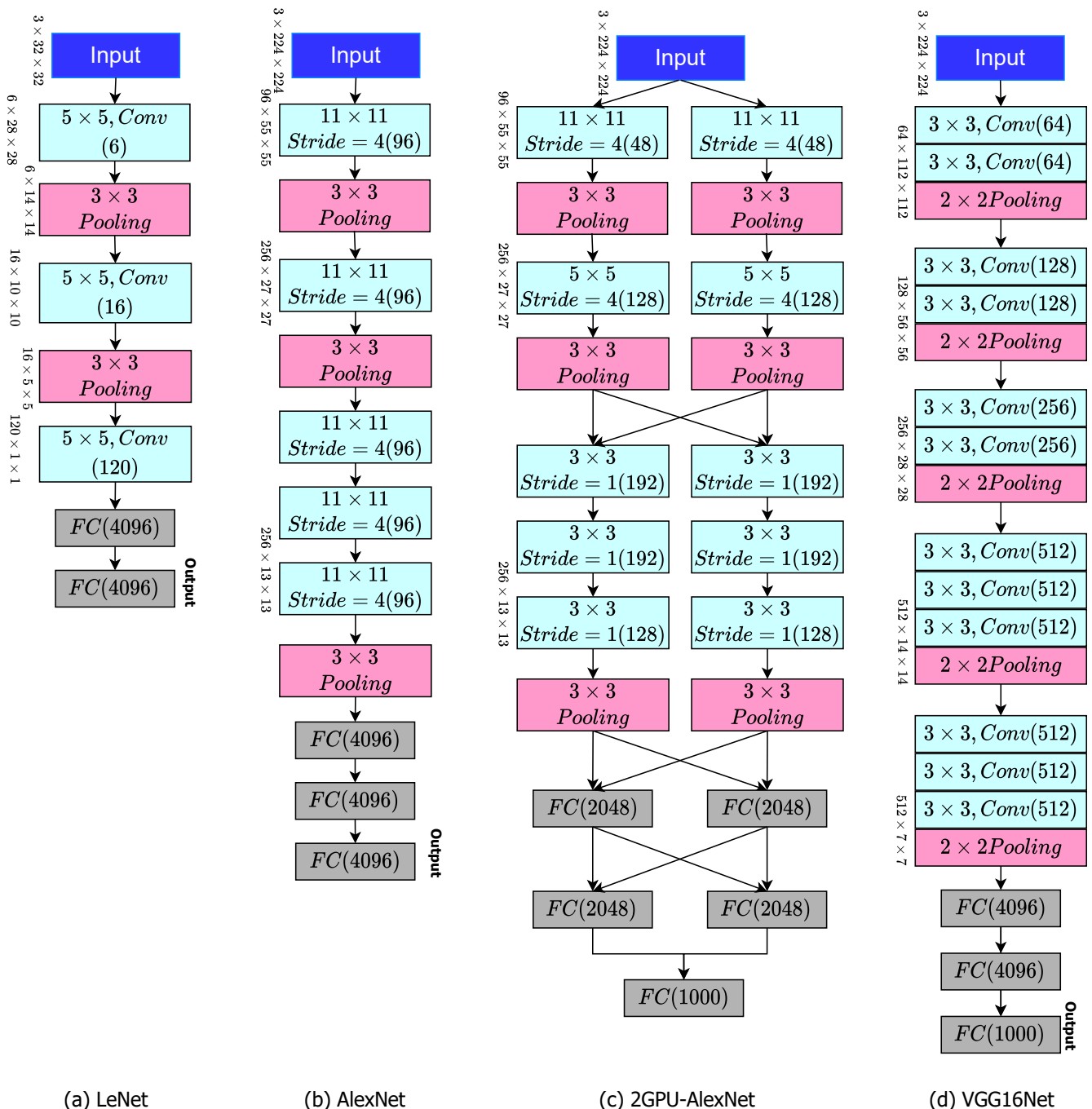

**Figure 2.** CNN Models: Architecture for LeNet, AlexNet, 2GPU-AlexNet, and VGG16Net.

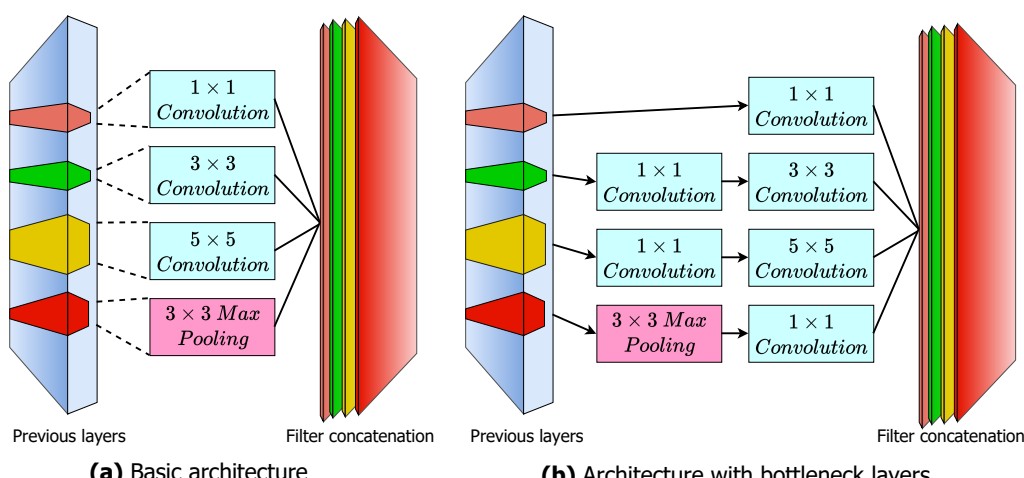

**Figure 3.** GoogleNet architecture.

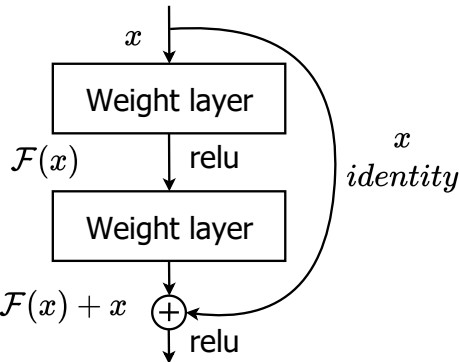

**Figure 4.** ResNet architecture.

*2.6. DenseNet*

DenseNet is a contemporary CNN architecture for visual object recognition that has achieved state-of-the-art performance while requiring fewer parameters. DenseNet is fairly similar to ResNet but for a few key differences. DenseNet uses its concatenates (.) attributes to mix the previous layer output with a future layer, whereas ResNet blends the previous layer with the future layers using an additive attribute (+). The training process of ResNet CNN becomes harder because of lack of gradient when deeper into the deeper features. This problem is resolved in DensNet by establishing a shortest path between one layer and its successor layer. This architecture establish shortest path between all layers, a architecture with $L$ layers establish $Z$ connections which is equal to $\frac{L(L+1)}{2}$. In this architecture, every layer carries a limited tuning parameters and for each layers are convoluted with 12 filters. Implicit deep supervision character improves the flow of the gradient through the network. The outcome features of all layers can directly pass though loss function and its gradients are available for all the layers as shown in Figure 5. In the architecture of DenseNet, each layer outputs $k$ feature maps, where $k$ is the growth factor. The bottleneck layer of $1 \times 1$ followed by $3 \times 3$ convolutions and $1 \times 1$ convolutions, output $4k$ feature maps, for ImageNet, the initial convolution layer, outputs $2k$ feature maps [16].

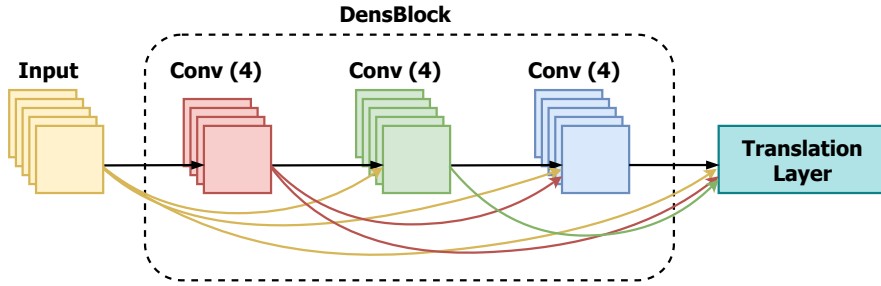

**Figure 5.** DenseNet architecture.

*2.7. MobileFaceNet*

This network architecture uses newly introduced convolution called depth-wise separable convolution which allows the network tune with less hyper-parameters as in Figure 6. It also provides flexibility in the selection of a right sized model dependence on the application of designers or users by introducing two simple global hyperparameters i.e., Width Multiplier (Thinner models) and Resolution Multiplier (Reduced representation). In standard convolution, the application of filters across all input channels and the combination of these values is done in a single step—whereas, in depthwise convolution, convolution is performed in two stages: depthwise convolution—filtering stage and pointwise convolution—combination stage. Efficiency has a trade-off with accuracy [17]. Depthwise convolution reduces the number of computations because of the fact that multiplication is an expansive operation compared to addition.

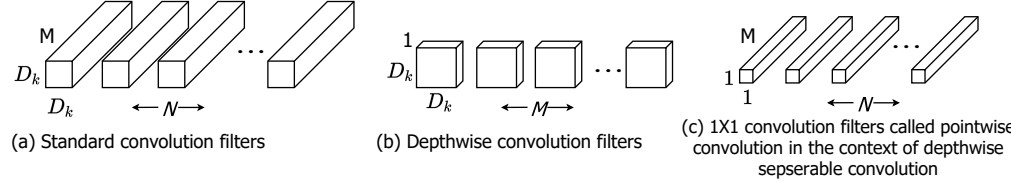

(a) Standard convolution filters     (b) Depthwise convolution filters     (c) 1X1 convolution filters called pointwise convolution in the context of depthwise sepserable convolution

**Figure 6.** Depthwise separable convolution.

*2.8. ShuffleNet*

The ShuffleNet has advantages over MobileNet, it has higher accuracy with less computational time and less hyper-parameters. It is used for many applications like cloud robotics, drones and mobile phones. It overcome the drawback of expansive point-wise convolution by introducing group convolution point-wise and to override side effects by introducing channel shuffler. Practical experiment shows that ShuffleNet has an accuracy of 7.8 percent higher than MobileNet and its computation time is much faster (13 times) than MobileNet. The group convolution already explained in ResNet or AlexNet [18]. Figure 7a is a bottleneck unit with depth-wise seperable convolution ($3 \times 3$ DWConv). Figure 7b is a ShuffleNet architecture with point-wise group convolution with channel shuffler. In figure, second level of group convolution in point-wise is to provide the shortest path for reduction of channel dimension. Figure 7c gives a ShuffleNet architecture including stride of size 2. The shufflent performance is better because of the group convolution and shuffling process in channel.

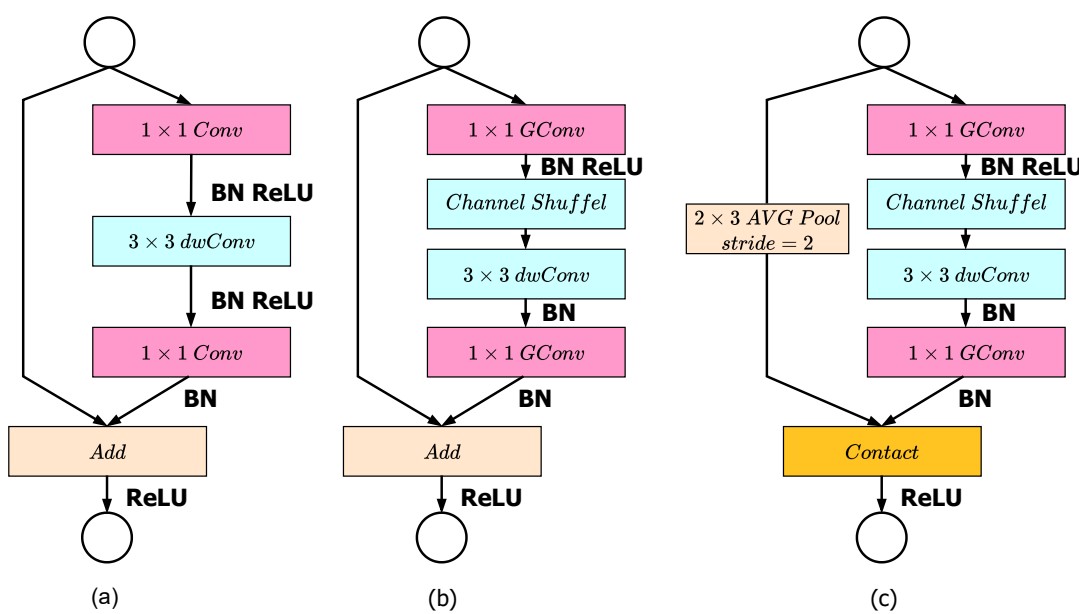

**Figure 7.** ShuffleNet architecture: (**a**) bottleneck unit with depthwise convolution (DWConv); (**b**) ShuffleNet unit with pointwise group convolution (GConv) and channel shuffle; (**c**) ShuffleNet unit with stride = 2.

### 2.9. EffNet

Unlike other previously discussed networks, spatial separable convolution was introduced in EffNet and is almost similar to depth-wise separable convolution which is used in MobileNet. In separable convolution, filter or kernel matrix *k* partitioned into two vector of size *k*1 and *k*2 (indirectly multiplication of matrix *k*1 and *k*2). This partition allows convolution of two one dimensional is equivalent to convolution of one two dimensional. For the sake of better understanding lets take s simple example,

$$\begin{pmatrix} 3 & 6 & 9 \\ 4 & 8 & 12 \\ 5 & 10 & 15 \end{pmatrix} = \begin{pmatrix} 3 \\ 4 \\ 5 \end{pmatrix} \begin{pmatrix} 1 & 2 & 3 \end{pmatrix}$$

$$k = k1 * k2$$

Now, perform convolution with *k*1 followed by convolution with *k*2 which leads reduction in number of multiplication. With this simple concept, the of multiplication required to perform convolution comes down to 6 (each one has 3 multiplication) in place of 9 multiplications. This reduction in multiplication leads to improvement in training speed and reduce the computational complexity of the network as in Figure 8. The major drawback with this process is that, possibility of portioning is not possible for all the kernels. This drawback is the bottleneck of Effnet particularly while in training process [19].

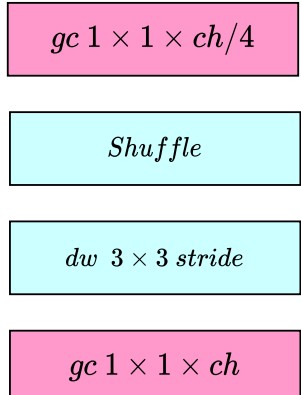

**Figure 8.** EffNet architecture.

### 3. Overview of the Encryption Algorithms under Study

In this section, we give an overview of the encryption algorithms that we will compare between them.

#### 3.1. DNA Algorithm

The confusion matrix generated from the chaotic crypto system is applied on input image, and these encrypted image pixels are further diffused in accordance with the transformation of nucleotides into respective base pairs. Images of any size to be encrypted are reshaped to specific sizes and rearranged as an array which follows a strategy generated by a chaotic logistic map. In the second stage, according to a DNA inbuilt process, the confused pixels are shuffled for further encryption. In the third stage, every DNA nucleotide is transformed to a respective base pair by means of repetitive calculations that follow Chebyshev's chaotic map [20].

#### 3.2. AES Algorithm

Here, images are transformed from gray-levels to respective strings; upon the converted strings, the AES algorithm performs image encryption. It is a symmetric key block cipher to safeguard the confidential information. AES is used to encrypt the images and was implemented on board a circuit or in a programmatic form in a computer. AES comes with three forms with key sizes of 128, 192, and 256, respectively. In the case of an AES-128 bit key, for both encryption and decryption of strings, it uses a key length of 128 bits. It is symmetric cryptography, also known as private key cryptography, and encryption and decryption uses the same key; therefore, the transmitter and recipient both should know and utilize a certain secret key. The sensitive and security levels can be protected with any key length. Key lengths of 192 or 256 bits are required for a high level of security. In this article, we used an AES-512 scheme for encryption of images; in addition, a bit wise shuffling was applied [21]. AES performance depends on key size and number of iterations chosen for shuffling and is proportional to size of the key and iterations. In the encryption stage, in general, 10, 12, or 14 iterations are used. Except for the last round, each round comprises four steps, shown in Algorithm 1:

---

**Algorithm 1:** AES encryption flow.

---

    1. KeyExpansion: Using the AES key schedule, round keys are produced from cipher keys. Each round of AES requires a distinct 128-bit round key block, plus one extra.

    2. SubBytes: By means of an "s-box" nonlinear lookup table, one byte is substituted with another byte [21].

    3. ShiftRows: Byte transposition occurs, one, two, or three bytes are used to cyclically change the state matrix's of the second, third, and fourth rows to the left.

    4. The final step is to find final state matrix calculation by multiplying fixed polynomial and current state matrixes.

---

### 3.3. Genetic Algorithm (GA)

The genetic algorithm encryption follows natural reproduction of genetics of humans or animals. In this, images are encrypted in three levels: first stage—reproduction, second stage—crossover, and third stage—mutation.

Reproduction: The production of new solution/encrypted image is obtained by fusion of two images (one original and another reference image); sometimes, it is called offspring and the algorithm security depends on the best selection of offspring/reference image. [22].

Crossover: The images to be encrypted are represented with eight bits per pixel; these eight bits of each pixel are equally portioned. For crossover operation, select any two pixels and its equivalent binary bits; now, swap the first four bits of one pixel with the last four bits of another pixel. The swapped pixel and its equivalent intensity generate a new value. This process is repeated for all the pixels in an image to get an encrypted image.

Mutation: This stage is not mandatory in general, but, for some reason, we perform this stage. It is simply a complement/invert (making 1 to 0 or 0 to 1) of any bit of any pixel, and its key depends on the position of the selected bit for complement.

### 3.4. Bit Slicing Algorithm

This image encryption scheme is simple, flexible, and faster, the security of encryption and randomness in genetic algorithm are improved with image bit slicing and rotating slices at any preferred angles 90, 180, and 270 degrees. The gray scale image is sliced into eight slices because it takes eight bits to represent any intensity [23]. The algorithm is given in Algorithm 2:

---

**Algorithm 2:** BSR image encryption flow.

---

    1. In an image, all pixels are represented with their respective 8-bit binary equivalents.

    2. Segregate the image into eight parts based on bits starting from MSB to LSB.

    3. For each slice, now apply rotation operation with predefined angles.

    4. Perform the three above for specified iterations and, after that, convert encrypted bits to gray scale intensity to get encrypted images.

---

### 3.5. Chaos Algorithm

The authors were at first enthusiastic in employing simple chaotic maps including the tent map and the logistic map since the quickness of the crypto algorithm is always a key aspect in evaluating the efficiency of a cryptography algorithm. However, new image encryption algorithms based on more sophisticated chaotic maps shown in 2006 and 2007 that using a higher-dimensional chaotic map could raise the effectiveness and security of crypto algorithms. To effectively apply chaos theory in encryption, chaotic maps must be built in such a way that randomness produced by the map may induce the requisite confusion matrix and diffusion matrix. In obtaining the confusion matrix, the pixel positions are changed in a specific fashion and ensure no change in pixel intensity levels.

Similarly, a diffusion matrix is obtained by modifying the pixel values sequentially in accordance with the order of sequence which are generated by chaotic crypto systems [24].

### 3.6. RSA Algorithm

Ron Rivest, Adi Shamir, and Leonard Adleman (RSA) created this algorithm back in 1977. The RSA algorithm is used to encrypt and decrypt the information. The public-key cryptosystems include the RSA cryptosystem. RSA is a cryptographic algorithm that is commonly used to transmit safe information. The RSA technique is used to encrypt the images, with two huge prime integers and a supplementary value chosen as the public key. The prime numbers are kept hidden from the public. The public key is used for image encryption, while the private key is used to decrypt them. It is not only used for image but also for text encryption. Consider two large prime numbers $r$ and $s$ as public and private keys respectively for encryption of images. Take two integers $f$ and $e$ in such a way that $f \times e \bmod \phi(m) = 1$. With these four selected integers, images are encrypted with the simple formula $D = q^f \bmod \phi(m)$, where $q$ is an original input image, $f$ is a publicly available key, and $\phi(m)$ is a product of ($r$-1) and ($s$-1) ensures that $gcd(f, \phi(m)) = 1$ i.e., $f$ and $\phi(m)$ are co-primes. $D$ is the encrypted image after encryption through RSA. To get back the original image, use $R = D^e \bmod \phi(m)$, where e is a key available only for private people [25] (Show in Algorithm 3).

---

**Algorithm 3:** Image encryption using RSA.

---

1. Initially access the original gray-scale image of any size ($R$)
2. Now, select two different large prime numbers $r$ and $s$.
3. Measure the m value which is equal to $m = rs$.
4. Now, calculate $\Phi(m) = \Phi(r)\Phi(s) = (r-1)(s-1) = m - (r+s-1)$, where function $\Phi$ is Euler's totient function.
5. Select another integer $f$ (public key) in such a way that $1 < f < \Phi(m)$ ; and ;
   $gcd(f, \Phi(m)) = 1$, in which $f$ and $\Phi(m)$ are co-primes.
6. Calculate e as $e = (f-1) \bmod \Phi(m)$; i.e., $e$ is the multiplicative modular inverse of $f$ ($modulo \Phi(m)$).
7. Get the encrypted image, $D = Sf \bmod \Phi(m)$.
8. For gray-scale images, perform $D = D \bmod 256$, since image contains the pixel intensity levels between 0 to 255.
9. To decrypt the encrypted image, perform, $S = De \bmod 256$.
10. Then, the original input image $R = S \bmod \Phi(n)$.

---

### 3.7. Rubik's Cube Principle (RCP)

This encryption scheme follows the strategy of Rubik's Cube Principle. In this, the gray-scale image pixels are encrypted by changing the position of the individual pixels, and this change follows the principle of Rubik's Cube. Initially, two keys are generated randomly; with these keys, image pixels are gone through the XOR operation bit by bit between odd row-wise and columns-wise. In the same way, image pixels are gone through the XOR operation bit by bit between even row-wise and columns-wise with flipped versions of two secret keys. This process is repeated until the maximum or termination criteria reached [26]. Algorithm 4 shows the detailed description of the RCP algorithm for image encryption.

---

**Algorithm 4:** Rubik's Cube flow.

---

1. Let us assume an image $I_o$ of size $M \times N$, and assume it is represented with $\alpha$-bits. Now, two randomly generated vectors $L_S$ and $L_D$ of size $M$ and $N$, respectively. Each elements in $L_S(i)$ and $L_D(j)$ can take a random number between a set A of range between 0 to $2\alpha - 1$.

2. Pre-define the maximum iterations count (itrmax), and set itr to zero.

3. For every iterations, itr is incremented by one: itr = itr + 1.

4. For every row of image $I_o$,

(a) calculate addition of all pixels in ith row, and is calculated by

$$\alpha(i) = \sum_{j=1}^{N} I_0(i,j), i = 1, 2, 3, 4, \ldots, M, \tag{1}$$

(b) Now, calculate $M\alpha(i)$ by doing modulo 2 of $\alpha$ (i),

(c) The ith row is shifted right or left or circular by $L_S(i)$ positions (pixels of images are moved to right or left direction by $K_R(i)$; after this operation, the first pixel becomes the last, and the last pixel becomes first), as per the following equation:

$$if \begin{cases} M\alpha(i) & = 0 \quad right \quad circular \quad shift \\ M\alpha(i) & \neq 0 \quad left \quad circular \quad shift \end{cases} \tag{2}$$

5. Similarly, for every column of image $I_o$,

(a) calculate addition of all pixels $\beta(i)$ in jth column, and is calculated by

$$\beta(i) = \sum_{i=1}^{N} I_0(i,j), i = 1, 2, 3, 4, 5\ldots, M, \tag{3}$$

(b) now, calculate $M\beta(j)$ by doing modulo 2 of $\beta(j)$.

(c) the jth column of image is shifted up or circular or down by $L_D(i)$ positions, by following equations:

$$if \begin{cases} M\beta(i) & = 0 \quad up \quad circular \quad shift \\ M\beta(i) & \neq 0 \quad down \quad circular \quad shift \end{cases} \tag{4}$$

These steps (4 & 5) create a scrambled image $I_{SCR}(i)$.

6. With the help of vector $L_D$, a bit-wise XOR operation is performed on each row of scrambled image $I_{SCR}(i)$ by means of following equation:

$$I_1(2i-1, j) = I_{SCR}(2i-1, j) \oplus L_D(j), \tag{5}$$

$$I_1(2i, j) = I_{SCR}(2i, j) \oplus rot180(L_D(j)). \tag{6}$$

In the above equation, $\oplus$ shows XOR operation (bit-wise) and rot180($L_D$) shows a flip operation on vector $K_C$ from right to left.

7. With the help of vector $L_S$, the XOR operation (bit-wise) is performed on the column of scrambled image by means of following equation:

$$I_{ENC}(i, 2j-1) = I_1(i, 2j-1) \oplus (L_S(j)), \tag{7}$$

$$I_{ENC}(i, 2j) = I_1(i, 2j)rot180 \oplus (L_S(j)), \tag{8}$$

where $rot180(L_S(j))$ shows the flip operation from left to right with respect to vector $K_R$.

8. Repeat step 1 to step 7 until itr = itrmax. 9. Finally, encrypted image $I_{ENC}$ is generated and process is terminated; otherwise, the algorithm moves to step 3.

---

### 3.8. Hill Cipher Algorithm

It also comes, under symmetric key encryption algorithm, in which one can retrieve the decryption key by means of very simple transformation or repetitive computations. Encryption of images with this is very simple, easy, and a limited time-consuming process. It just replaced the original image pixels of size $m$ with the Hill cipher image pixels of the same size. Let us take a simple example, let us assume three pixels with names as $Q_1, Q_2, Q_3$, and assume the Hill cipher algorithm and replace them with $C_1, C_2, C_3$, as per the following procedure:

$$C_1 = (L_{11}Q_1 + L_{12}Q_2 + L_{13}Q_3)mod26 \tag{9}$$

$$C_2 = (L_{21}Q_1 + L_{22}Q_2 + L_{23}Q_3)mod26 \tag{10}$$

$$C_3 = (L_{31}Q_1 + L_{32}Q_2 + L_{33}Q_3)mod26 \tag{11}$$

This can be represented by matrices as follows:

$$\begin{bmatrix} C_1 \\ C_2 \\ C_3 \end{bmatrix} = \begin{bmatrix} L_{11} & L_{12} & L_{13} \\ L_{21} & L_{22} & L_{23} \\ L_{31} & L_{32} & L_{33} \end{bmatrix} \begin{bmatrix} Q_1 \\ Q_2 \\ Q_3 \end{bmatrix} \tag{12}$$

The above can also be represented as: $C = LQ$, where $Q$ is plain pixels and $C$ is Hill cipher pixels. $L$ is a key matrix that performs the transformation from original to hill cipher and is reversible. To get back the the original image, simply use $Q = L^{-1}C$. In this algorithm, the generation of reversible key matrix handles complex computations and is very hard task [27].

## 4. Security Analysis of the Studied Encryption Algorithms

To compare the studied encryption algorithms, we conceived experiments illustrating the effect of the image encryption on robot facial recognition and security issues in cloud environments. These experiments also aim at measuring the time needed to recognize faces in cloud and robotic environments. Our analysis of the results led us to compare the accuracy of the robot's facial recognition on two levels:

- facial recognition of the robot. This includes the cloud and excluding the cloud. It should be noted that the recognition algorithm is executed in the on-board robot;
- the effect of the encryption algorithms on the accuracy of the robot's facial recognition.

We took a dataset composed of $92 \times 112$ grayscale pgm images to assess the performance of the encryption algorithms [28]. For each of the 40 different subjects, 10 separate images were provided. To ensure some variability in the image acquisition conditions, images were taken at different times with varying lighting, for some subject. Moreover, the images were taken for different facial expressions (eyes open/closed, smiling/not smiling), different details of the face (glasses/no glasses), and poses of the head with tilts and rotations up to 20 degrees. We took a dark background when capturing all the images. For each of the 40 subjects, four images with different poses are used as the training dataset, while the remaining six images for each are used as the test database. This leads to 160 images for training and 240 images for testing. Likewise, experiments were carried out by varying the number of training images per face. In addition, we have undertaken robot face recognition experiments. To do this, we encrypted training and test images with the algorithms mentioned above.

### 4.1. Parameters Used for Security Analysis

Attacks such as the Statistical Attack, Plain Text Attack, Cipher Text Attack, Differential Attack, and Brute Force Attack present serious practical difficulties when implementing the algorithms operating them [29]. In this article, we used the following security measure parameters.

### 4.1.1. Histogram

The histogram is a graph indicating the number of repetitions of each of the gray levels when going through all the pixels of the given image. On the *x*-axis, we find the gray levels from 0 to 255, as shown in Figure 9 for the input image. Certainly, when the histogram of the original image is significantly different from that of the encrypted image, better security is ensured. It is expected that the histogram of the encrypted image is uniformly distributed. In this case, it becomes difficult to divide the content of the original image [30].

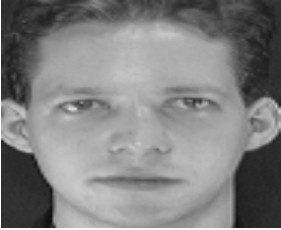 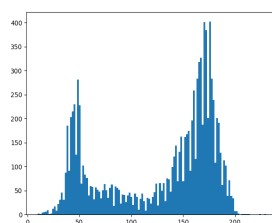 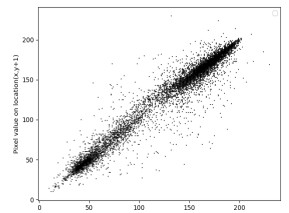 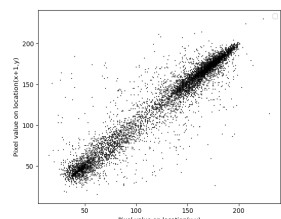

**Figure 9.** From left to right: Input image, Histogram, Horizontal scatter plot and Vertical scatter plot

### 4.1.2. Correlation Coefficient (CC)

This coefficient gives us a quantitative indication of the similarity between the original image and the encrypted one [31]. It is calculated as follows:

$$C_{xy} = \frac{cov(x,y)}{\sqrt{D(x)}\sqrt{D(y)}} \tag{13}$$

The variables *x* and *y* respectively denote the gray values of two adjacent horizontal and vertical, or diagonal pixels in the given image. We apply the following formulas:

$$cov(x,y) = \frac{1}{N}\sum_{i=1}^{N}(x_i - E(x))(y_i - E(y)) \tag{14}$$

$$E(x) = \frac{1}{N}\sum_{i=1}^{N}x_i, \quad E(y) = \frac{1}{N}\sum_{i=1}^{N}y_i \tag{15}$$

$$D(x) = \frac{1}{N}\sum_{i=1}^{N}(x_i - E(x))^2 \quad D(y) = \frac{1}{N}\sum_{i=1}^{N}(y_i - E(y))^2 \tag{16}$$

where *N* stands for the number of used pairs of pixels.

### 4.1.3. Scatter Plot

It is a graphical representation of the correlation between two adjacent pixels. While the original image gives us a perfect positive correlation, the encrypted image generally shows no correlation. When the plot is a straight line, this means that the correlation between the two measured pixels is high. If the image contains random or quasi-random values, no correlation appears, as depicted in Figure 9.

### 4.1.4. Number of Pixels Change Rate (NPCR)

This parameter provides a quantitative indication for the pixel change rates in the encrypted image when a pixel change is introduced at the original image [32]. Let us consider the input image *T*1 and its encrypted version *T*2, where *T*1 has only one pixel difference. The NPCR metric of two images is defined as follows:

$$NPCR = \frac{\sum_{i,j} s(i,j)}{M * N} * 100 \tag{17}$$

where $s(i, j) = \begin{cases} 0 & if \quad T1(i,j) = T2(i,j) \\ 1 & if \qquad otherwise \end{cases}$

Here, $M$ is the number of rows in the given image, $N$ is the number of columns in this image, and $(i,j)$ represent the position of one pixel.

### 4.1.5. Unified Averaged Changed Intensity (UACI)

This parameter determines the average intensity difference between original image and its encrypted version [31]:

$$UACI = \frac{1}{M*N} \left[ \sum_{i,j} \frac{s(i,j)}{255} \right] * 100 \tag{18}$$

### 4.1.6. Mean Square Error (MSE)

The mean square error (MSE) is the measurement of difference between the original and cipher images [33]. The high value of MSE is related to a high amount of difference between original image and cipher image. It can be calculated by the following equation:

$$MSE = \frac{1}{M*N} \sum_{i=0}^{M-1} \sum_{j=0}^{N-1} [I(i,j) - K(i,j)]^2 \tag{19}$$

where $I$ and $K$ represent the plain image and the encrypted image.

### 4.1.7. Peak Signal-to-Noise Ratio (PSNR)

The peak signal-to-noise ratio (PSNR) measures the conformity between the plain and cipher images [33]:

$$PSNR = 10log\frac{255^2}{MSE}(db) \tag{20}$$

where MSE is the Mean Square Error.

### 4.2. Security Analysis and Discussion

To compare the studied encryption algorithms, we proceed as follows. After choosing a grayscale image from the given database, we apply the encryption algorithms to measure the security parameters and the recognition accuracy in the cloud and robot environments. Figure 9 shows the test image as well as the corresponding histogram and scatter plot. It turned out that the correlation coefficient of the test image is slightly smaller than 1, and more precisely equal to 0.955. Table 1 shows the security parameters of the studied encryption algorithms. Moreover, Figure 10 shows the histogram and the horizontal and vertical scatter point of the encrypted image, for the studied encryption algorithms. It turns out that, if the histogram of the encrypted image is uniform, then it becomes very difficult to break the security key. Indeed, the image with uniform values does not present relevant information. For the range of algorithms used here and the histogram of the image encrypted with Hill Cypher, the Rubik's Cube and DNA methods are found to be more consistent. Therefore, they are more resistant to statistical analysis attacks compared to other methods. It is worth noting that an encryption algorithm is more secure when the correlation between neighboring pixels inside the encrypted image is low. According to Table 1, the correlation coefficient of the encrypted image is almost zero. By applying the DNA method, we obtained a very small correlation coefficient, which is almost zero. In Table 2, we see that the encryption is fast for the chaos algorithm, while the encryption time is bigger for the AES. It should be noted that we carried out our experiments in robot (desktop) and cloud environments. We used the specifications of Table 3. It is clear from the Table 1 that the DNA method produces the highest NPCR value, whereas the GA method gives the highest UACI value. In addition, the GA method turned out to be producing the lowest peak signal-to-noise ratio (PSNR) [34] value. All our observations converge on the

fact that, with the exception of bit slicing and GA, the various algorithms generally offer a relatively high level of security for the encrypted image.

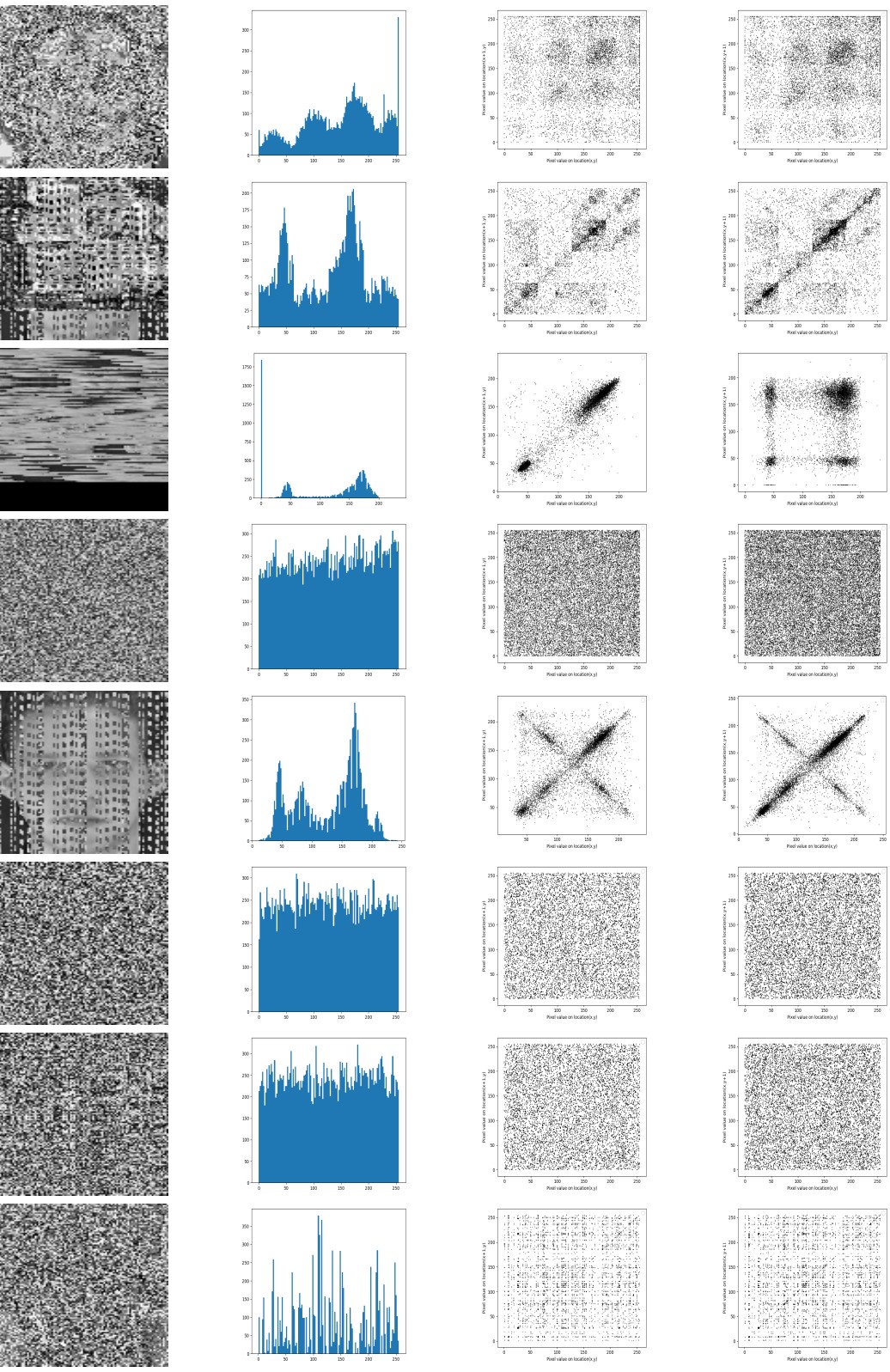

**Figure 10.** Security measures with proposed encryption algorithms. From left to right: encrypted image histogram, horizontal, vertical scatter plot. From top to bottom: AES, Bit-slice, Chaos, DNA, GA, Hill cypher, RCP, and RSA.

**Table 1.** Security measures of the studied encryption algorithms.

| Algorithm | NPCR | CC | MSE | PSNR | UACI |
|---|---|---|---|---|---|
| DNA | 99.446 | **0.0615** | 6372.06 | 10.088 | 24.6696 |
| AES | 99.572 | 0.1188 | 7033.57 | 9.6590 | 26.5178 |
| GA | **34.297** | 0.3828 | **4293.95** | **11.802** | **9.7257** |
| Bit slicing | 86.403 | 0.4029 | 6483.72 | 10.0125 | 22.4563 |
| Chaos | 99.116 | 0.9529 | 7262.96 | 9.5196 | 26.2900 |
| RSA | 99.996 | 0.0678 | 7432.85 | 9.4192 | 29.8552 |
| RCP | 99.621 | 0.0285 | 7542.40 | 9.3557 | 29.4176 |
| Hill-Cypher | 99.631 | 0.0328 | 7541.06 | 9.3564 | 29.4601 |

**Table 2.** Encryption time, average, and standard deviation to encrypt 40 images with various encryption algorithms.

| Algorithm | Time | Avg. time | Std. Time |
|---|---|---|---|
| DNA | 4.8877 | 4.9674 | 0.89898 |
| AES | 131.11 | 66.092 | 37.7128 |
| GA | 1.4454 | 1.4376 | 0.00937 |
| Bit slicing | 2.3361 | 2.2671 | 0.02760 |
| Chaos | **0.0478** | **0.0473** | **0.00097** |
| RSA | 1.4402 | 1.4357 | 0.03938 |
| RCP | 1.2341 | 1.3879 | 0.98765 |
| Hill-Cypher | 2.8552 | 1.4336 | 0.82310 |

**Table 3.** Specifications of cloud and robot.

| | Cloud | Robot |
|---|---|---|
| Processor | Intel(R) Xeon(R) Silver 4114 CPU @ 2.20GHz | Intel core(R) i7-9700k CPU @ 3.60GHz |
| RAM | 40 GB | 32 GB |
| OS | 64-bit Linux | 64-bit Windows |

## 5. Performance Analysis of CNN Based Robot Face Recognition Models

In this section, we present the results of the performance of the previously discussed CNN models and the impact of the encryption algorithms on the accuracy of these models. For each CNN model, we measure its accuracy first with clear images and then with encrypted images using one of the studied encryption algorithms. Moreover, we have also evaluated the performance in robot and cloud environment.

To study the impact of the encryption algorithms on the accuracy of the studied CNN models, we proceed as follows. First, we encrypt the images, then the studied CNN model is trained using these encrypted images, and finally, the CNN model is tested using a subset of the encrypted images and consequently, the accuracy is measured.

We studied the performance of the previously nine CNN models namely, Lenet, AlexNet, ResNet, VGG16Net, GoogleNet, DenseNet, MobileFaceNet, ShuffleNet, and EffNEt.

During the pre-processing step, the dataset images are resized to a size suitable for each CNN model and gray scale images are converted to RGB images.

Experimental results are conducted using the ORL database [35]. The Olivetti Research Laboratory (ORL) face dataset contains a set of face images.

### 5.1. Simulation Settings

In the NVIDIA GEFORCE GTX 1050TI variant, all tests were carried out utilizing the Windows platform with Intel core(R) i7-9700k CPU @ 3.60 GHz with 32 GB RAM in the case of robot environment and Intel(R) Xeon(R) Silver 4114 CPU @ 2.20 GHz with 32 GB RAM in the case of a cloud environment. The Python version 3.7.3 with TENSORFLOW

version $< 1.15.0 >= 1.14.0$ is introduced to test the process and extract the features and for classification of images. As mentioned above, a pre-processing is needed before the formation process is started for the convolution neural network architectures. A re-scale is used for the resized images of data set in order to convert the images to $224 \times 224$ as AlexNet input and to $224 \times 224$ as ResNet input. Based on established Quality parameters, the efficiency of the trained neural network is assessed based on accuracy.

*5.2. Learning Rate*

One of the important hyperparameter of CNN models is the learning rate. The learning rate measures the efficiency of the CNN network weights and more precisely how well are adapted to the loss rate. The lower the rating, the smoother the downhill direction is. Although it could be a smart strategy for us to ensure we do not skip any local minimas (through a low learning rate), it may also imply that we take a long time to converge—especially if we remain on a plateau region. In addition, the learning rate affects how fast our model can converge to a local minimum (the best possible accuracy is achieved). To have it right will mean that we would have less time to learn the algorithm. Given the increasing sophistication of the facial recognition network model, an effective learning rate is especially challenging to achieve, since the scale of various parameters differs considerably and may be changed during the training cycle, which requires a long time.

Thus, experiments are conducted to find the suitable learning rate for each CNN model. Figures 11 and 12 show the learning rate versus accuracy of GoogleNet and Resnet, respectively. For GoogleNet, accuracy is maximum at a value of 0.0005 of learning rate. Regarding ResNet, accuracy is maximum at a value of 0.0007 of learning rate. For other CNN models a learning rate value of 0.0001 was used during experiments.

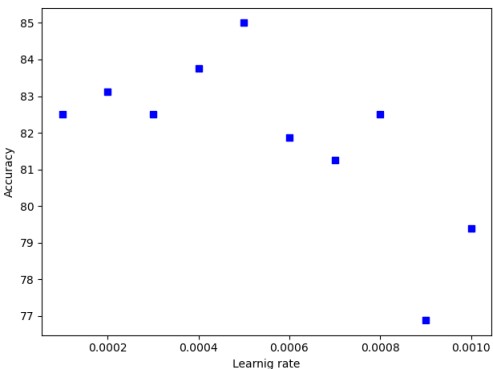

**Figure 11.** GoogleNet Learning rate.

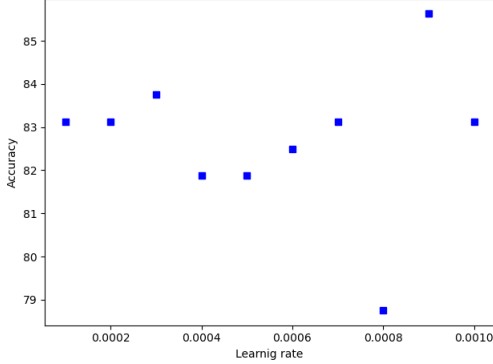

**Figure 12.** ResNet Learning rate.

*5.3. Accuracy*

The accuracy of the studied CNN models depends on the efficient feature extraction from encrypted images. Although a CNN model can easily extract features from clear images, it is a challenging task for it to extract features from encrypted images.

The studied CNN models use different filters in convolution layers, pooling layers, and followed different strategies for features extraction, which leads to different accuracy of face recognition. Moreover, images with different poses, illumination effects, and disguise might lead to low face recognition accuracy value.

Regarding encryption algorithms, they introduce some distortion in the images features. Thus, image features are distorted after encryption and consequently CNN models can lead to different results. Moreover, among the studied encryption algorithms some are linear and some are nonlinear. Therefore, they have different effect on the accuracy of the CNN models.

According to the results shown in Table 4 the **Bit Slice, Chacos** and **GA** encryption algorithms preserve the features after encryption, and therefore, give better accuracy compared to other encryption algorithms. More precisely, for the studied CNN based face recognition models, their accuracy obtained using these encrypted algorithms is close to the accuracy obtained using clear images.

Table 4 shows the accuracy obtained with various algorithms including PCA. It It is clearly noted from the table that the face recognition accuracy of CNN models is better than accuracy of PCA for clear images and encrypted images. From the studied CNN models, Effnet gives the best accuracy and outperforms other models with all encrypted algorithms. Compared to PCA, EffNet improves accuracy by 16.2%, 6.16%, 20.4%, 22.1%, 279.88%, 9.56%, 392.1%, 366.32%, and 480 % for plain image, AES, Bit slice, Chacos, DNA, GA, Hill, RSA and RCP, respectively. Additionally, Tables 5 and 6 show the testing and training time of various CNN models. The testing time of all CNN models for one image is less than one second, and training time is 6 to 8 minutes for the used ORL database. Figures 13–17 show the loss and accuracy curves of LeNet, AlexNet, Vgg16Net, ResNet, and DenseNet, respectively. To train the CNN model, Softmax loss function is used in this paper. It is a cascaded combination of softmax function, the last fully connected layer of corresponding CNN model, and cross-entropy loss. In training CNN models, weights $wk$ (where k is number of training class, here 40) of connected layers are normalized to scale parameter $s$, and, in the same way, outcome features of last convolution layers amplitudes are normalized to $s$ [36]. As a result, softmax loss is calculated from the below equation for a given input vector feature $x$, and, assuming respective ground truth label $y$, then

$$L_1 = -log \frac{e^{scos(\theta_{wy},x)}}{e^{scos(\theta_{wy},x)} + \sum_{k \neq y}^{k} e^{scos(\theta_{wy},x)}} \tag{21}$$

where $cos(\theta_{w_y}, x) = w_k^T x$ is the cosine similarity and $\theta_{w_y}, x$ is the angle between $w_k$ and $x$. The learned features with softmax loss are prone to be separable, rather than to be discriminative for robot face recognition. In addition, the time needed to produce recognition results (with LeNet) for one image for the cloud turned out to be smaller than that required for the robot as depicted in Table 7.

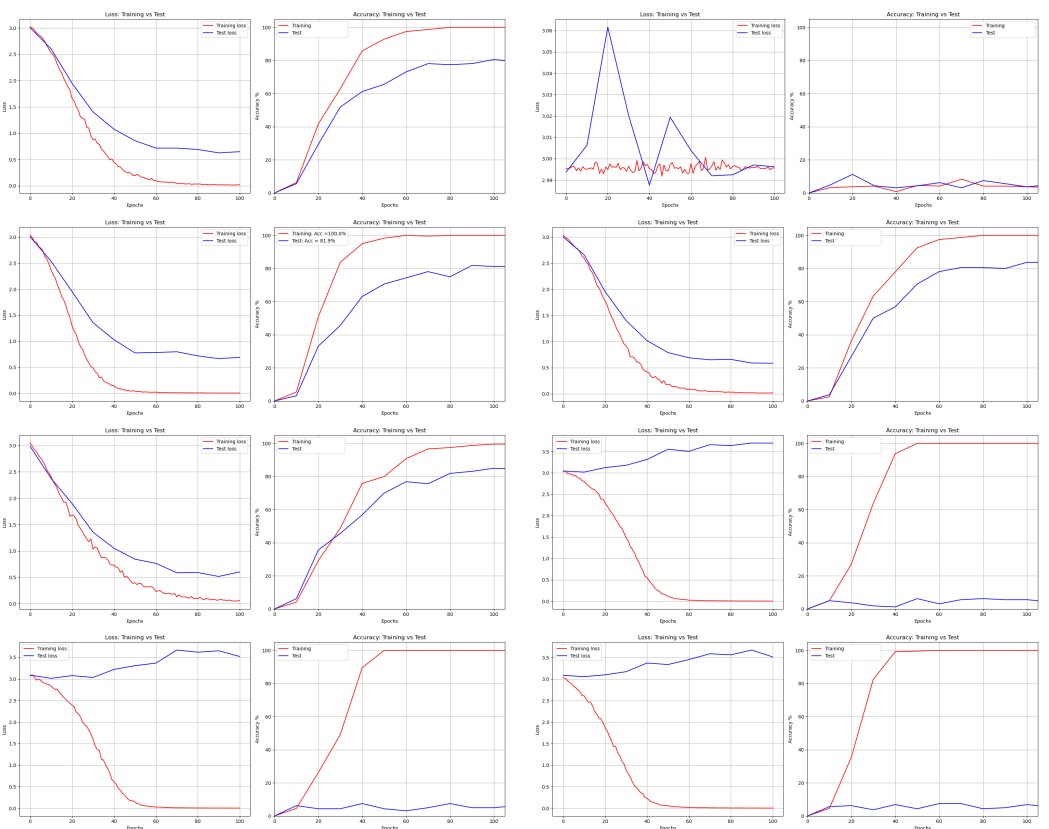

**Figure 13.** Loss and Accuracy of **AlexNet** from left to right and top to bottom: DNA, AES, Bit Slice, Chaco, GA, Hill, RCP, and RSA.

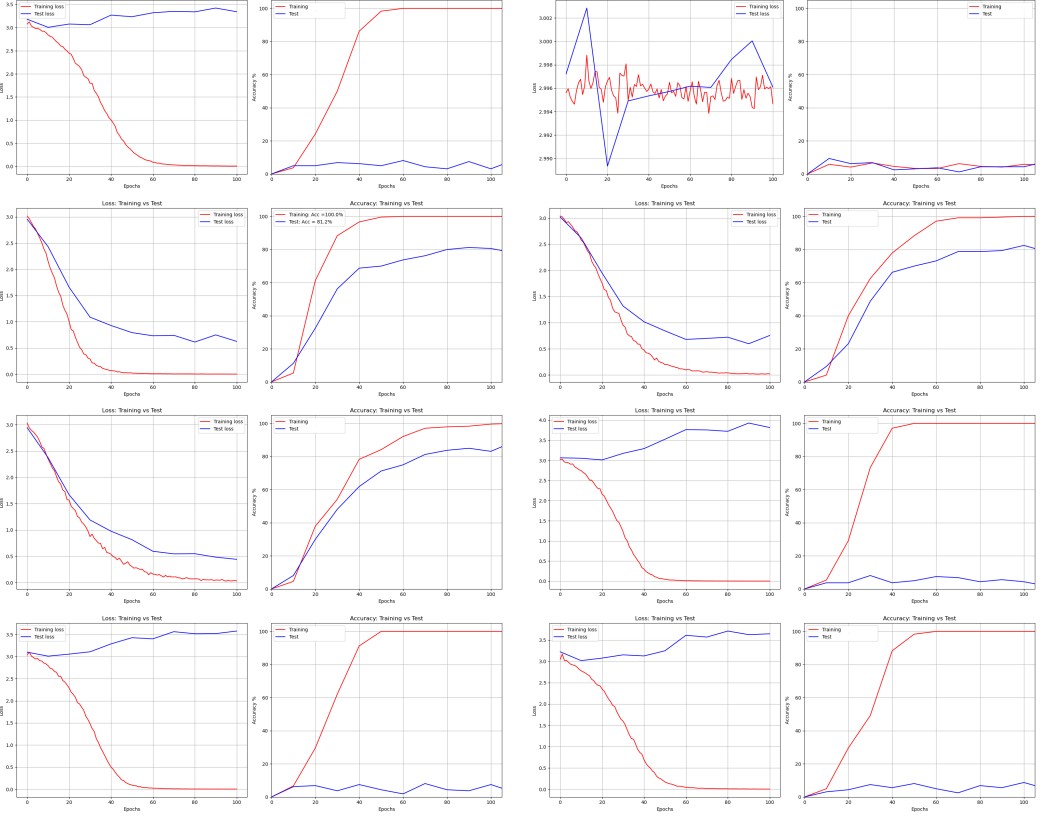

**Figure 14.** Loss and Accuracy of **Vgg16Net** from left to right and top to bottom: DNA, AES, Bit Slice, Chaco, GA, Hill, RCP, and RSA.

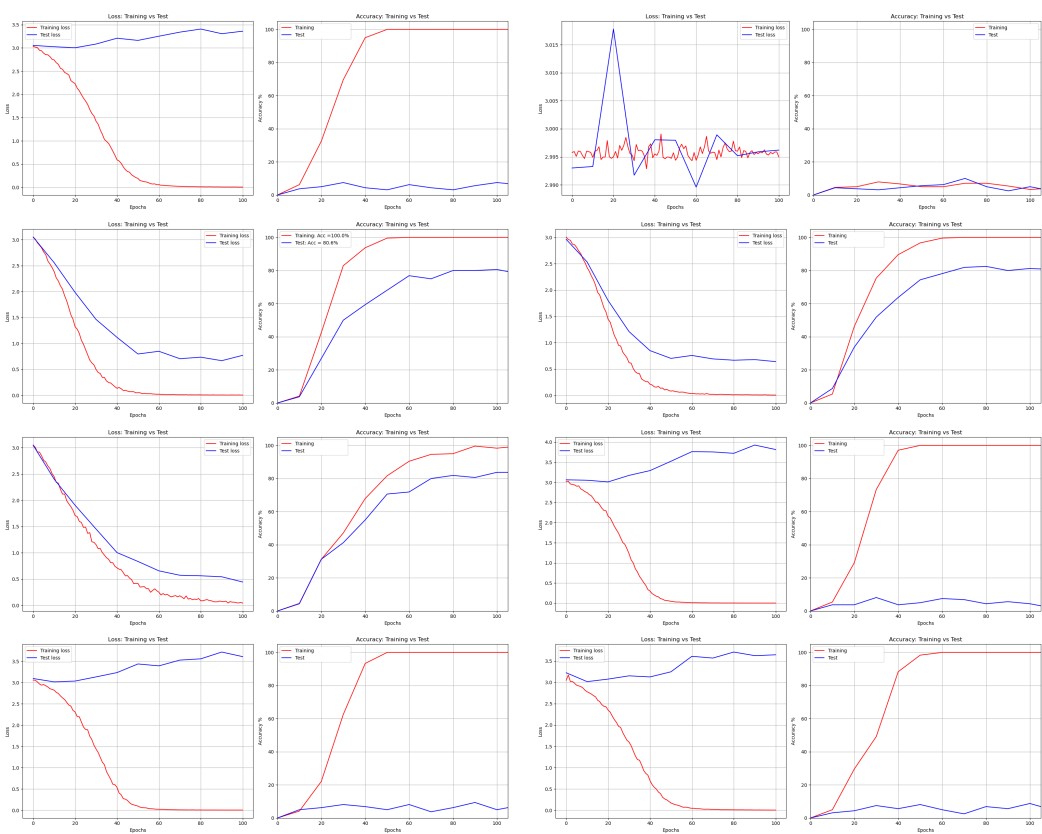

**Figure 15.** Loss and Accuracy of **GoogleNet** from left to right and top to bottom: DNA, AES, Bit Slice, Chaco, GA, Hill, RCP, and RSA.

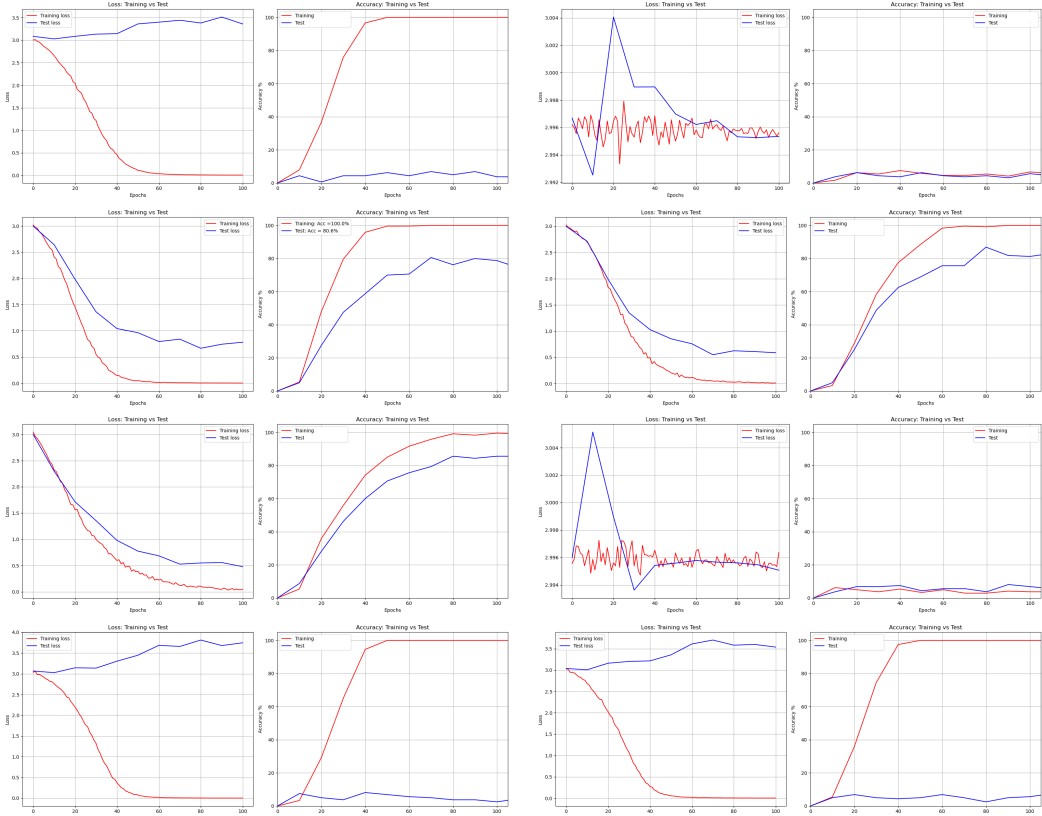

**Figure 16.** Loss and Accuracy of **ResNet** from left to right and top to bottom: DNA, AES, Bit Slice, Chaco, GA, Hill, RCP, and RSA.

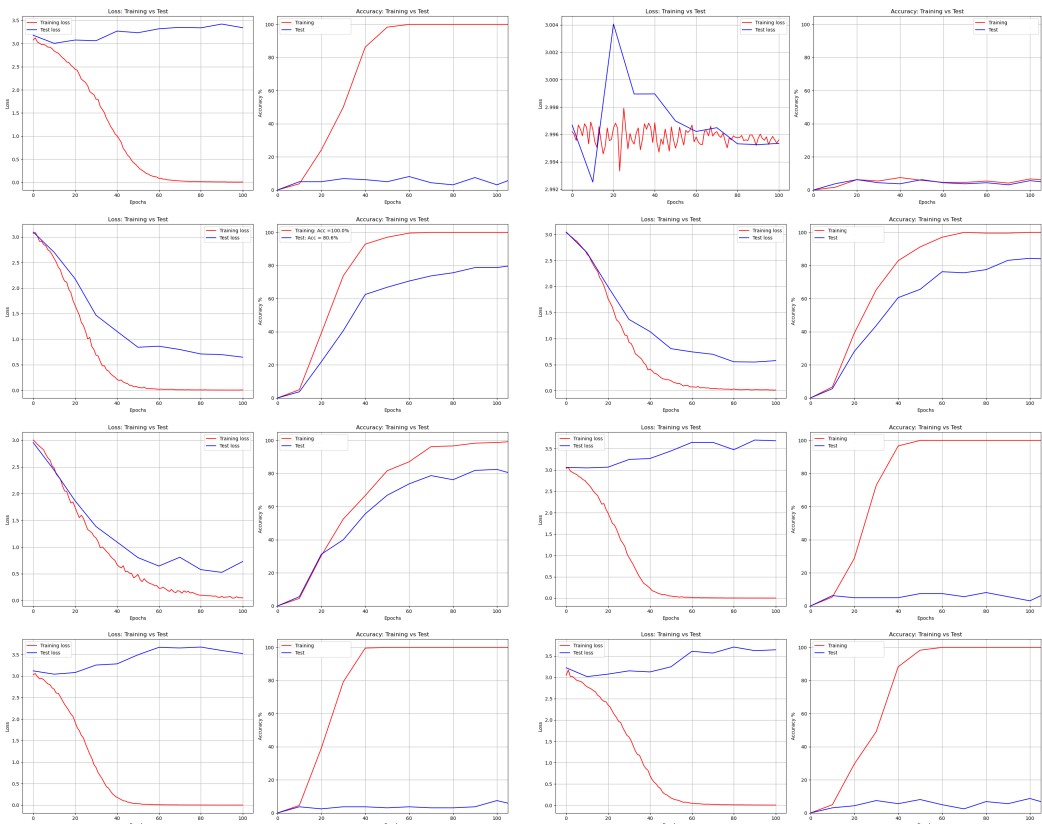

**Figure 17.** Loss and Accuracy of **DenseNet** from left to right and top to bottom: DNA, AES, Bit Slice, Chaco, GA, Hill, RCP, and RSA.

**Table 4.** Accuracy of the studied CNN models.

| CNN Model | Without Enc. | AES | RCP | Hill | RSA | DNA | Bit Slice | Chacos | GA |
|---|---|---|---|---|---|---|---|---|---|
| PCA | 80.4 [37] | 2.50 | 2.51 | 2.91 | 2.91 | 3.33 | 70.0 | 72.50 | 80.4 |
| LeNet | 82.5 [38] | 6.25 | 7.50 | 8.75 | 9.38 | 6.25 | 80.62 | 84.38 | 82.50 |
| Alexnet | 84.3 [39] | 6.88 | 7.50 | 6.25 | 8.75 | 6.88 | 82.50 | 81.88 | 85.00 |
| VGG16Net | 85.6 [13] | 7.50 | 8.12 | 9.38 | 8.12 | 6.88 | 80.62 | 86.87 | 85.62 |
| GoogLeNet | 85.7 [14] | 8.12 | 9.38 | 6.88 | 8.75 | 8.24 | 81.25 | 82.50 | 88.75 |
| ResNet | 85.3 [39] | 9.38 | 10.00 | 8.12 | 8.75 | 7.50 | 80.62 | 82.50 | 87.50 |
| DenseNet | 87.9 [40] | 7.50 | 11.25 | 8.12 | 6.88 | 6.25 | 80.62 | 83.13 | 85.00 |
| MobileFaceNet | 88.6 [41] | 11.7 | 12.34 | 11.34 | 9.25 | 10.54 | 81.35 | 86.25 | 87.92 |
| ShuffleNet | 90.3 [41] | 13.2 | 15.34 | 12.31 | 11.2 | 11.21 | 83.34 | 87.32 | 87.91 |
| EffNet | 93.5 [19] | 14.5 | 17.98 | 14.32 | 13.57 | 12.65 | **84.34** | **88.54** | **88.09** |

**Table 5.** Training time (Sec) of CNN models on robots.

| CNN Model | Without Enc. | AES | Bit Slice | Chacos | DNA | GA | Hill | RSA | RCP |
|---|---|---|---|---|---|---|---|---|---|
| LeNet | 450.84 | 439.51 | 458.51 | 431.033 | 443.16 | 441.45 | 452.97 | 453.86 | 435.61 |
| Alexnet | 441.77 | 449.66 | 435.14 | 451.181 | 430.95 | 438.31 | 431.39 | 432.91 | 454.7 |
| VGG16Net | 449.45 | 436.39 | 451.82 | 454.15 | 457.70 | 453.40 | 457.56 | 455.43 | 454.68 |
| GoogLeNet | 449.17 | 435.62 | 457.40 | 455.15 | 450.76 | 448.60 | 452.73 | 455.01 | 457.59 |
| ResNet | 444.69 | 434.37 | 457.75 | 451.39 | 467.23 | 448.57 | 448.06 | 457.42 | 458.59 |
| DenseNet | 443.37 | 433.25 | 458.75 | 459.0 | 457.64 | 457.62 | 454.84 | 458.95 | 458.01 |
| MobileFaceNet | 449.39 | 440.73 | 462.67 | 455.76 | 455.12 | 461.85 | 450.90 | 451.40 | 452.65 |
| ShuffleNet | 453.77 | 458.78 | 449.67 | 431.071 | 455.47 | 458.02 | 450.36 | 452.73 | 452.29 |
| EffNet | 454.44 | 457.17 | 433.81 | 457.401 | 448.97 | 432.93 | 438.35 | 446.41 | 458.73 |

**Table 6.** Testing time (Sec) of CNN models on robots.

| CNN Models | Without Enc. | AES | Bit Slice | Chacos | DNA | GA | Hill | RSA | RCP |
|---|---|---|---|---|---|---|---|---|---|
| LeNet | 0.559 | 0.749 | 0.98 | 0.67 | 0.793 | 0.612 | 0.876 | 0.628 | 0.753 |
| Alexnet | 0.85 | 0.945 | 0.98 | 0.774 | 0.569 | 0.575 | 0.629 | 0.92 | 0.627 |
| VGG16Net | 0.671 | 0.671 | 0.765 | 0.796 | 0.656 | 0.812 | 0.640 | 0.796 | 0.656 |
| GoogLeNet | 0.687 | 0.753 | 0.781 | 0.656 | 0.671 | 0.562 | 0.968 | 0.812 | 0.578 |
| ResNet | 0.538 | 0.765 | 0.640 | 0.968 | 0.906 | 0.906 | 0.671 | 0.875 | 0.906 |
| DenseNet | 0.527 | 0.875 | 0.671 | 0.781 | 0.890 | 1.000 | 0.875 | 0.781 | 0.781 |
| MobileFaceNet | 0.765 | 0.828 | 0.796 | 0.828 | 0.718 | 0.906 | 0.781 | 0.781 | 0.687 |
| ShuffleNet | 0.907 | 0.622 | 0.965 | 0.675 | 0.598 | 0.626 | 0.808 | 0.737 | 0.676 |
| EffNet | 0.915 | 0.793 | 0.775 | 0.959 | 0.643 | 0.879 | 0.877 | 0.69 | 0.784 |

**Table 7.** Computational time to recognition one image in cloud and robots.

| Source | Time of Execution in Seconds |
|---|---|
| Robot | 0.559 |
| Cloud | 0.014 |

## 6. Conclusions

We studied several approaches for robot secure face recognition in the cloud environment. We used different algorithms to encrypt a set of images. The encrypted versions were used for training and testing. We trained the robot with various deep learning algorithms. We adopted some tests to measure safety, time complexity, and recognition accuracy. Both cloud and robot environments were considered. The findings showed that some algorithms were well suited for the security criterion. Others were better placed for recognition accuracy. The findings also revealed that, compared to a large range of algorithms, the genetic algorithm is a good candidate for better security and better recognition accuracy. Our study also revealed that the recognition accuracy provided by AES, RSA, and RCP is not reasonably high, although these methods are more secure. The percentages of improvement from PCA to EffNet were 16.2%, 6.16%, 20.4%, 22.1%, 9.56%, 279.88%, 392.1%, 366.32%, and 480% for plain images and encrypted images using AES, Bit slice, Chacos, GA, DNA, Hill, RSA, and RCP encryption algorithms, respectively. By comparing cloud and robot environments, we concluded that the recognition was faster for the cloud. It is advisable to run the same algorithm on the cloud to improve the robot's recognition speed.

**Author Contributions:** Conceptualization, C.K. and O.C.; methodology, C.K. and O.C.; writing—original draft preparation, C.K.; writing—review and editing, C.K. and O.C. and A.H.; visualization, A.H.; supervision, H.H.; project administration, O.C.; funding acquisition, A.Z. All authors have read and agreed to the published version of the manuscript.

**Funding:** This research was funded by Taif university Researchers Supporting Project Number (TURSP-2020/114), Taif University, Taif, Saudi Arabia.

**Institutional Review Board Statement:** Not applicable

**Informed Consent Statement:** Not applicable.

**Data Availability Statement:** Publicly available dataset were analyzed in this study. This data can be found here: [https://www.kaggle.com/tavarez/the-orl-database-for-training-and-testing].

**Acknowledgments:** The authors thank Taif University Research Supporting Project Number (TURSP-2020/114), Taif University, Taif, Saudi Arabia. The authors would like to thank also Asma Cheikhrouhou from University of Carthage, Tunis, Tunisia, for her English editing services.

**Conflicts of Interest:** The authors declare no conflict of interest.

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
