# Peer review of "Privacy Preserving Face Recognition in Cloud Robotics: A Comparative Study"

_applsci, doi:10.3390/app11146522_

Round 1
Reviewer 1 Report
The authors proposed a method to improve speed , accuracy and security in robot face recognition task. They proposed a deep learning architecture. Full description and experimental results benchmarking are reported.
The paper is well written and the pipeline sounds good. It has been accepted after a minor spell check.
Author Response
Thank you very much for the positive feedback. In this revision, we have checked and fixed the linguistic and grammatical mistakes. We have also rephrased several sentences to enhance the quality of our article.
Reviewer 2 Report
The authors addressed a very interesting problem and performed a lot of analysis. However, the article needs a huge amount of editing and polishing of text which is not in the scope of reviewing an article. There is also a design fault in the machine learning work where the authors do not have a validation set which could lead to overfitting of machine learning models. The present structure of the article is more like a thesis not like an article.
Author Response
The authors really appreciate the reviewer's comments. In this revision, we have focused on incorporating the provided suggestions and further reorganizing the article.
Comment1: The article needs a huge amount of editing and polishing of text which is not in the scope of reviewing an article.
Answer1: Thank you for mentioning this issue. We have carefully proofread the paper and it has a better quality now.
Comment2: There is also a design fault in the machine learning work where the authors do not have a validation set which could lead to overfitting of machine learning models.
Answer2: In this paper, we have used the ORL dataset, among 400 images 10 percentage i.e 40 images are considered for validation and rest for training.
Comment3:The present structure of the article is more like a thesis not like an article.
Answer3: Thank you for mentioning this issue. The structure of the paper was updated and improved.
Reviewer 3 Report
In the paper, the Authors propose an improvement of speed and accuracy for mainly face recognition in encrypted domain without loosing privacy of robot. Some important experiments were illustrated for robot face recognition through various deep learning algorithms. However, there are few necessary improvements, as below.
- Some CNNs extract features from images before the main classification. I haven’t noticed such solution in this paper.
- In Table I, there are metrics that weren’t earlier defined.
- The term “References” is needed. All formulas should be indexed. The paragraph “In training …” (line 440) is supposed to be divided.
Author Response
We are very thankful for your comments and suggestions which helped us to improve the quality of our revised manuscript. According to your suggestions, we have modified the relevant parts and answered each of your comments as below.
Comment1: Some CNNs extract features from images before the main classification. I haven’t noticed such solution in this paper.
Answer1: The aim of this paper is to study the accuracy of the CNNs models and the impact of encryption on this accuracy.
In CNN, classification follows feature extraction and extracted features are feed to classification, the studied CNNs all follow the same strategy.
Comment2: In Table I, there are metrics that weren’t earlier defined.
Answer2: Thank you for mentioning this issue. All table 1 metrics are now defined.
Comment3: The term “References” is needed. All formulas should be indexed. The paragraph “In training …” (line 440) is supposed to be divided.
Answer3:
- The title of the "References" section was added.
- All formulas are now indexed.
- The paragraph "In formation ..." (line 440) is now in a separate paragraph